# Action Space Design in Reinforcement Learning for Robot Motor Skills

**Julian Eßer**[*1], **Gabriel B. Margolis**[*2], **Oliver Urbann**[1], **Sören Kerner**[1], **Pulkit Agrawal**[2]
[1] Fraunhofer Institute for Material Flow and Logistics IML
[2] Massachusetts Institute of Technology

**Abstract:** Practitioners often rely on intuition to select action spaces for learning. The choice can substantially impact final performance even when choosing among configuration-space representations such as joint position, velocity, and torque commands. We examine action space selection considering a wheeled-legged robot, a quadruped robot, and a simulated suite of locomotion, manipulation, and control tasks. We analyze the mechanisms by which action space can improve performance and conclude that the action space can influence learning performance substantially in a task-dependent way. Moreover, we find that much of the practical impact of action space selection on learning dynamics can be explained by improved policy initialization and behavior between timesteps.

**Keywords:** Reinforcement Learning, Action Spaces, Sim-to-Real

## 1 Introduction

Reinforcement learning (RL) is a powerful tool for synthesizing robot motor skills. However, practitioners must carefully select the action space where learning occurs, a decision often guided by intuition. For instance, a wheeled robot might be associated with a wheel velocity action space, a legged robot with joint positions, and a manipulator with Cartesian space targets. For some well-studied tasks, the field has converged on common action spaces. For example, position control action spaces are widely adopted for learning legged locomotion [1, 2, 3]. However, to actuate the robot's motion, these position commands must be converted into torques by a feedback law, both to perform simulation and control the real robot. This raises several questions: What properties of position control make it particularly useful for legged locomotion tasks instead of directly learning to act with torques? Is position control helpful for all types of robot tasks, or are there other, potentially more effective, action spaces for systems with different dynamics? How might other tasks that have not been extensively studied benefit from different action space design choices?

The historical motivations for considering joint position or velocity control instead of torque differ from their roles as action spaces in reinforcement learning. Classical systems aim for precise regulation to simplify and stabilize high-level control tasks. In contrast, RL aims to maximize cumulative rewards over time, which might involve deliberately altering position and velocity tracking errors to achieve the overall objective. The key point raised originally by Hwangbo et al. [1] is that the position target in RL is state-indexed and not time-indexed as typical in non-learning approaches. Still, a difference often arises in performance among action spaces due to their impact on the system dynamics and the learning dynamics. Just as PD controllers support high-bandwidth control beneath a slower model-based optimization [4, 5], a learned policy might benefit from a high-frequency low-level control loop, achieving better performance at a lower frequency. On the other hand, the action space may also experience benefits in RL that are separate from those in model-based control. For example, the action space can alter the policy's exploratory behavior or architecture-related biases.

---

*Denotes equal contribution. ✉: `julian.esser@iml.fraunhofer.de`, `gmargo@mit.edu`

8th Conference on Robot Learning (CoRL 2024), Munich, Germany.

| Task | Default Action Space | | |
|---|---|---|---|
| | Position | Delta Position | Torque |
| AllegroHand | ✓ | | |
| Ant | | | ✓ |
| Anymal | | ✓ | |
| AnymalTerrain | ✓ | | |
| BallBalance | | ✓ | |
| Cartpole | | | ✓ |
| FrankaCabinet | | ✓ | |
| Humanoid | | | ✓ |
| ShadowHand | ✓ | | |

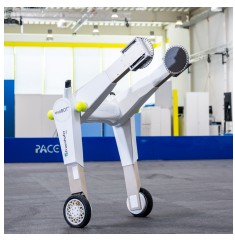
(a) evoBOT

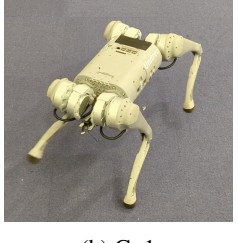
(b) Go1

Figure 1: **Diverse action spaces** in the OmniIsaacGymEnvs suite (left) are present across multiple robotic tasks. We aim to guide practitioners in selecting action space for new tasks by analyzing a subset of the suite (highlighted) and the evoBOT and Go1 platforms (right) as a case study.

**Contributions**   Through this lens, we examine the contribution of action space selection on the Unitree Go1 quadruped, the evoBOT hybrid robot [6], and the OmniIsaacGymEnvs task suite [7]. We pose four novel questions that haven't been explored in prior work:

1. *How does the choice of action space impact learning in systems with distinct dynamics?* We find that even for the same locomotion objective, two robots with distinct dynamics have opposite trends with regard to action space selection. Section 4.1

2. *Is action space selection mostly about tuning exploration?* We find that for some tasks, the performance gap between action spaces can be mostly recovered by intervening in the initial exploration behavior, while for others, this aspect is less influential. Section 4.2

3. *Is action space selection mostly about expressive capacity?* We find that despite a significant gap in reinforcement learning performance between action spaces, performant policies can be re-expressed in all those spaces with nearly equal performance. Section 4.3

4. *How important is the policy behavior between timesteps?* We evaluate the impact of high-level and low-level control frequency across action space representations and find that the tuning of these parameters is coupled. Section 4.4

## 2   Related Work

**Action Spaces**   Robots operate within a physics-defined action space, where for instance electric actuators use current to track torque via a high-frequency control loop [4]. While reinforcement learning policies can directly output torques [8, 9], many studies suggest alternative action spaces such as joint position [10, 2, 3], joint velocity [11, 6], or task-space setpoints [12], which are transformed to torques through feedback laws. Research indicates that action space choices significantly affect robot learning in various contexts, including character animation [13, 14], manipulation [15, 16], and flying robots [17]. While position control is often favored, some studies suggest joint velocity may be more effective for specific tasks. The OmniIsaacGymEnvs suite [7] underscores the complexity of action space implementation, featuring a balanced mix of action types. Our work focuses on configuration-space action spaces due to their relevance in sim-to-real RL.

**Inductive Bias through Action Space Design**   Action space selection serves as an inductive bias in reinforcement learning, influencing the policy's hypothesis space before learning starts [14]. This can also be viewed as a form of environment shaping [18] or a method of policy initialization [19].

**Temporal Aspects in Action Space Design**   Parameters like control frequency can impact learning as well; lower frequencies may simplify learning by shortening trajectories [20]. Additionally, gain choices for setpoint tracking influence performance and sim-to-real transfer [21, 22, 23]. Techniques like action filtering [24] and chunking [25] also can enhance efficiency in long-horizon tasks.

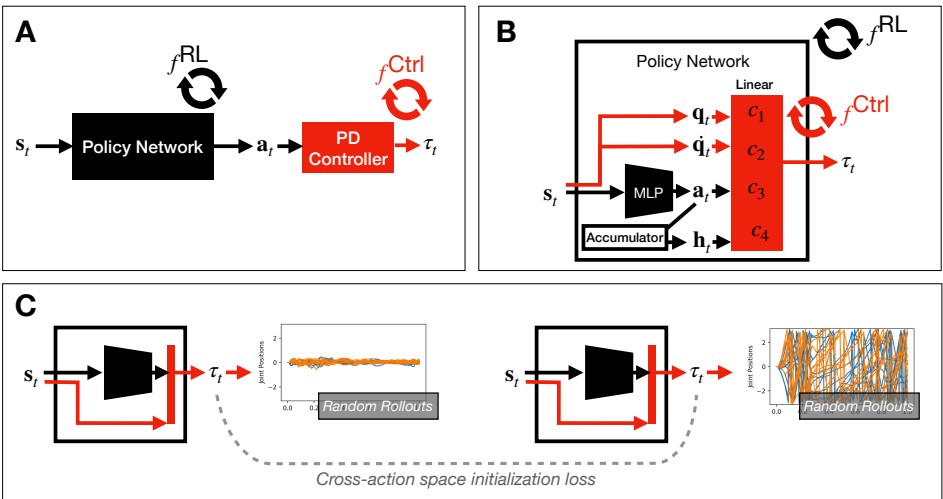

Figure 2: **Action Space Illustrative Diagram. A.** Depicts the classical decomposition of the policy network and low-level controller. **B.** Shows our generalized parameterization of action spaces, where the low-level controller acts as a linear layer when the learning frequency matches the control frequency. **C.** Suggests viewing differences among action spaces in terms of policy architecture and initialization, proposing pre-training one action space to replicate another's exploration.

## 3 Method

### 3.1 A Generalized Parameterization of Action Space

In a generalized form for RL, the transformation $\mathbf{T}$ from the policy output $\mathbf{a}_t$, state $\mathbf{s}_t$, and historical information $\mathbf{h}_t := \sum_{k=[0..t-1]} \mathbf{a}_k$ into motor torque commands $\tau_t$ can be expressed as:

$$\tau_t = \mathbf{T}(\mathbf{a}_t, \mathbf{s}_t, \mathbf{h}_t)$$

This formulation includes task-space action spaces as well as configuration-space action spaces. Here we focus on configuration space control where actions are expressed in terms of the joint position $\mathbf{q}_t$, velocity $\dot{\mathbf{q}}_t$, and torque $\tau_t$. Table 1 lists a set of common action spaces that fall in this category. If the command update and torque update operate at the same frequency (i.e. $f^{\text{Ctrl}} = f^{\text{RL}}$), we can combine them to express any of these action spaces as a linear mapping from the state and action to a torque. This leads us to the following general form for configuration-space action spaces, which is a special case of the formulation above:

$$\tau_t = \mathbf{T_J}(\mathbf{a}_t, \mathbf{s}_t, \mathbf{h}_t),$$
$$\mathbf{T_J}(\mathbf{a}_t, \mathbf{s}_t, \mathbf{h}_t) = c_1 \mathbf{q}_t + c_2 \dot{\mathbf{q}}_t + c_3 \mathbf{a}_t + c_4 \mathbf{h}_t.$$

We enumerate action spaces represented by this parameterization in Table 1. Each action space is simply a linear function of the state and action, independentlyly for each joint. Under this parameterization, we can see some simple relationships among the action spaces that may be unintuitive. Delta position control with a one-step integrator is equivalent to torque control with a damping term. Delta velocity control with a one-step integrator and torque control are also equivalent.

Figure 2 illustrates how the parameters $[c_1, c_2, c_3, c_4]$ may be interpreted as a final linear layer $\phi_A(\cdot)$ of the policy network $\pi(s_t)$. The choice among action spaces corresponds to a choice of manual initialization for these policy weights. This interpretation encourages us to consider whether the benefit of the action space is *architectural*, i.e. allowing the policy to learn parameters of this form results in better performance, or *initialization-related*, i.e. proper initialization of the network parameters seeds good performance.

| Action Space | Command Update | Torque Update | $[c_1 \quad c_2 \quad c_3 \quad c_4]$ |
|---|---|---|---|
| *Position* | $\mathbf{q}_t^{\text{des}} \leftarrow \mathbf{a}_t$ | $\tau_t \leftarrow K_p(\mathbf{q}_t^{\text{des}} - \mathbf{q}_t) - K_d\dot{\mathbf{q}}_t$ | $[-K_p \quad -K_d \quad K_p \quad 0]$ |
| *Delta Position (MS)* | $\mathbf{q}_t^{\text{des}} \leftarrow \mathbf{a}_t + \mathbf{q}_{t-1}^{\text{des}}$ | $\tau_t \leftarrow K_p(\mathbf{q}_t^{\text{des}} - \mathbf{q}_t) - K_d\dot{\mathbf{q}}_t$ | $[-K_p \quad -K_d \quad 0 \quad K_p]$ |
| *Delta Position (OS)* | $\mathbf{q}_t^{\text{des}} \leftarrow \mathbf{a}_t + \mathbf{q}_t$ | $\tau_t \leftarrow K_p(\mathbf{q}_t^{\text{des}} - \mathbf{q}_t) - K_d\dot{\mathbf{q}}_t$ | $[0 \quad -K_d \quad K_p \quad 0]$ |
| *Velocity* | $\dot{\mathbf{q}}_t^{\text{des}} \leftarrow \mathbf{a}_t$ | $\tau_t \leftarrow K_d(\dot{\mathbf{q}}_t^{\text{des}} - \dot{\mathbf{q}}_t)$ | $[0 \quad -K_d \quad K_d \quad 0]$ |
| *Delta Velocity (MS)* | $\dot{\mathbf{q}}_t^{\text{des}} \leftarrow \mathbf{a}_t + \dot{\mathbf{q}}_{t-1}^{\text{des}}$ | $\tau_t \leftarrow K_d(\dot{\mathbf{q}}_t^{\text{des}} - \dot{\mathbf{q}}_t)$ | $[0 \quad -K_d \quad 0 \quad K_d]$ |
| *Delta Velocity (OS)* | $\dot{\mathbf{q}}_t^{\text{des}} \leftarrow \mathbf{a}_t + \dot{\mathbf{q}}_t$ | $\tau_t \leftarrow K_d(\dot{\mathbf{q}}_t^{\text{des}} - \dot{\mathbf{q}}_t)$ | $[0 \quad 0 \quad K_d \quad 0]$ |
| *Torque* | $\tau_t \leftarrow \mathbf{a}_t$ | $-$ | $[0 \quad 0 \quad 1 \quad 0]$ |

Table 1: **Seven special cases** for the generalized parametrization of action spaces. The command update relates the policy output $\mathbf{a}_t$ to the desired state ($\mathbf{q}_t^{\text{des}}$, $\dot{\mathbf{q}}_t^{\text{des}}$, or $\tau_t$). The torque update relates the command to the desired state. MS = Multi-step, i.e. the action is applied as a delta to the previous desired state (e.g. $\mathbf{q}_{t-1}^{\text{des}}$) which is equal to the sum of all previous actions $\mathbf{h}_t$; OS = One-step, i.e. the action is applied as a delta to the current state.

## 3.2 Temporal Aspects of Action Space Design

The generalized parameterization of action spaces indicates they are equivalent to a linear function of state and policy output (see Fig. 2). However, this does not account for policy behavior between timesteps when the linear function operates at a higher frequency than policy evaluation, which is common in real systems. Higher-frequency modules can help bridge the sim-to-real gap, particularly in tasks where torque control is less important. For instance, a well-tuned low-level position controller can mitigate the need for accurate torque modeling. In force-sensitive applications like locomotion, successful methods achieve a low torque sim-to-real gap through direct drives or learning-based system identification. For further analysis, we denote the following frequencies:

**Physics Frequency ($f^{\text{Phys}}$):** The frequency of the physics step in simulation, ideally as high as possible, but balanced against computation speed. This frequency is crucial for simulating dynamics accurately while managing resource constraints in real systems.

**Control Frequency ($f^{\text{Ctrl}}$):** The frequency of torque updates (see Table 1), determines how frequently actions can change. A lower control frequency can limit the variability of action sequences but may enhance learning stability by shortening trajectories, thus simplifying credit attribution during training. We adopt the methodology from [20] to evaluate different control frequencies while adjusting the training parameters to keep the effective discount factor and batch size constant.

**Learning Frequency ($f^{\text{RL}}$):** The frequency of command updates (see Table 1), which, when lower than the control frequency, allows the same action to persist over several timesteps. This can improve credit attribution and speed up learning by reducing the number of policy evaluations, even as torque updates occur multiple times based on local state information.

## 3.3 Experimental Setup

**Policy Training** We utilize the Proximal Policy Optimization (PPO) [26] reinforcement learning algorithm, known for its effectiveness in sim-to-real motor control [1, 27]. Details on policy training and network architecture are provided in the appendix. We train on five continuous control tasks, including three from the OmniIsaacGymEnvs suite (BallBalance, Cartpole, FrankaCabinet) [7] and two custom locomotion tasks for evoBOT and Go1. The main focus is on evoBOT and Go1, where the reward combines velocity tracking and regularization terms. Training parameters and reward formulations are adapted from previous works [3, 6], with all terms independent of action representations for fair comparison. Simulation is conducted using NVIDIA Isaac Sim.

### (a) evoBOT

| | $c_3=$ | 10 | 15 | 20 | 30 |
|---|---|---|---|---|---|
| $c_1=-40$ | $c_2=-3.0$ | 0.47±0.00 | 0.50±0.01 | 0.53±0.00 | 0.54±0.01 |
| $c_1=-40$ | $c_2=-2.0$ | 0.48±0.00 | 0.50±0.00 | 0.53±0.00 | 0.56±0.01 |
| $c_1=-40$ | $c_2=-1.0$ | 0.49±0.00 | 0.51±0.01 | 0.56±0.01 | 0.68±0.01 |
| $c_1=-40$ | $c_2=-0.5$ | 0.51±0.01 | 0.53±0.00 | 0.63±0.02 | 0.61±0.01 |
| $c_1=-40$ | $c_2=0.0$ | 0.47±0.01 | 0.50±0.02 | 0.49±0.02 | 0.51±0.02 |
| $c_1=-30$ | $c_2=-3.0$ | 0.49±0.01 | 0.51±0.01 | 0.53±0.01 | 0.54±0.00 |
| $c_1=-30$ | $c_2=-2.0$ | 0.50±0.00 | 0.53±0.00 | 0.55±0.01 | 0.57±0.01 |
| $c_1=-30$ | $c_2=-1.0$ | 0.51±0.00 | 0.55±0.00 | 0.57±0.01 | 0.62±0.02 |
| $c_1=-30$ | $c_2=-0.5$ | 0.51±0.01 | 0.57±0.01 | 0.59±0.01 | 0.60±0.01 |
| $c_1=-30$ | $c_2=0.0$ | 0.49±0.00 | 0.50±0.02 | 0.48±0.02 | 0.52±0.01 |
| $c_1=-20$ | $c_2=-3.0$ | 0.51±0.01 | 0.53±0.00 | 0.54±0.00 | 0.56±0.00 |
| $c_1=-20$ | $c_2=-2.0$ | 0.51±0.00 | 0.53±0.01 | 0.54±0.01 | 0.59±0.01 |
| $c_1=-20$ | $c_2=-1.0$ | 0.52±0.01 | 0.56±0.01 | 0.58±0.01 | 0.63±0.00 |
| $c_1=-20$ | $c_2=-0.5$ | 0.53±0.00 | 0.57±0.00 | 0.59±0.01 | 0.62±0.00 |
| $c_1=-20$ | $c_2=0.0$ | 0.53±0.00 | 0.59±0.02 | 0.59±0.02 | 0.59±0.01 |
| $c_1=-10$ | $c_2=-3.0$ | 0.54±0.01 | 0.55±0.01 | 0.55±0.00 | 0.57±0.01 |
| $c_1=-10$ | $c_2=-2.0$ | 0.53±0.01 | 0.54±0.00 | 0.54±0.00 | 0.59±0.01 |
| $c_1=-10$ | $c_2=-1.0$ | 0.52±0.00 | 0.54±0.01 | 0.56±0.02 | 0.64±0.01 |
| $c_1=-10$ | $c_2=-0.5$ | 0.53±0.01 | 0.57±0.00 | 0.63±0.00 | 0.65±0.01 |
| $c_1=-10$ | $c_2=0.0$ | 0.55±0.00 | 0.60±0.01 | 0.62±0.01 | 0.63±0.02 |
| $c_1=0$ | $c_2=-3.0$ | 0.71±0.01 | 0.78±0.01 | 0.84±0.01 | 0.94±0.00 |
| $c_1=0$ | $c_2=-2.0$ | 0.76±0.02 | 0.88±0.01 | 0.94±0.02 | 0.97±0.03 |
| $c_1=0$ | $c_2=-1.0$ | 0.94±0.01 | 0.99±0.00 | 1.00±0.00 | 0.99±0.01 |
| $c_1=0$ | $c_2=-0.5$ | 1.00±0.00 | 0.99±0.01 | 1.00±0.00 | 0.97±0.03 |
| $c_1=0$ | $c_2=0.0$ | 0.99±0.01 | 1.00±0.00 | 0.99±0.01 | 0.98±0.01 |

### (b) Go1

| | $c_3=$ | 10 | 15 | 20 | 30 |
|---|---|---|---|---|---|
| $c_1=-40$ | $c_2=-3.0$ | 0.22±0.01 | 0.26±0.01 | 0.39±0.00 | 0.36±0.01 |
| $c_1=-40$ | $c_2=-2.0$ | 0.54±0.03 | 0.62±0.01 | 0.64±0.01 | 0.58±0.02 |
| $c_1=-40$ | $c_2=-1.0$ | 0.91±0.00 | 0.96±0.00 | 0.96±0.02 | 0.85±0.01 |
| $c_1=-40$ | $c_2=-0.5$ | 0.94±0.01 | 0.99±0.01 | 0.84±0.02 | 0.71±0.02 |
| $c_1=-40$ | $c_2=0.0$ | 0.91±0.01 | 0.84±0.02 | 0.46±0.06 | 0.27±0.05 |
| $c_1=-30$ | $c_2=-3.0$ | 0.17±0.01 | 0.27±0.01 | 0.39±0.00 | 0.43±0.09 |
| $c_1=-30$ | $c_2=-2.0$ | 0.55±0.01 | 0.64±0.00 | 0.64±0.00 | 0.57±0.07 |
| $c_1=-30$ | $c_2=-1.0$ | 0.96±0.01 | 0.99±0.01 | 0.97±0.01 | 0.80±0.03 |
| $c_1=-30$ | $c_2=-0.5$ | 0.98±0.01 | 0.97±0.01 | 0.77±0.02 | 0.66±0.05 |
| $c_1=-30$ | $c_2=0.0$ | 0.94±0.00 | 0.76±0.05 | 0.51±0.04 | 0.30±0.15 |
| $c_1=-20$ | $c_2=-3.0$ | 0.16±0.03 | 0.34±0.02 | 0.44±0.06 | 0.65±0.04 |
| $c_1=-20$ | $c_2=-2.0$ | 0.47±0.01 | 0.65±0.01 | 0.64±0.00 | 0.78±0.00 |
| $c_1=-20$ | $c_2=-1.0$ | 0.95±0.01 | 1.00±0.01 | 0.96±0.00 | 0.68±0.02 |
| $c_1=-20$ | $c_2=-0.5$ | 1.00±0.01 | 0.94±0.02 | 0.78±0.03 | 0.60±0.00 |
| $c_1=-20$ | $c_2=0.0$ | 0.95±0.01 | 0.71±0.02 | 0.52±0.09 | 0.52±0.05 |
| $c_1=-10$ | $c_2=-3.0$ | 0.04±0.03 | 0.32±0.17 | 0.40±0.18 | 0.69±0.02 |
| $c_1=-10$ | $c_2=-2.0$ | 0.62±0.01 | 0.70±0.04 | 0.67±0.03 | 0.67±0.01 |
| $c_1=-10$ | $c_2=-1.0$ | 0.95±0.01 | 0.82±0.03 | 0.73±0.02 | 0.62±0.03 |
| $c_1=-10$ | $c_2=-0.5$ | 0.96±0.04 | 0.74±0.01 | 0.64±0.02 | 0.53±0.05 |
| $c_1=-10$ | $c_2=0.0$ | 0.72±0.02 | 0.62±0.01 | 0.57±0.01 | 0.43±0.06 |
| $c_1=0$ | $c_2=-3.0$ | 0.52±0.02 | 0.59±0.02 | 0.61±0.07 | 0.48±0.04 |
| $c_1=0$ | $c_2=-2.0$ | 0.61±0.01 | 0.59±0.02 | 0.54±0.03 | 0.51±0.02 |
| $c_1=0$ | $c_2=-1.0$ | 0.56±0.03 | 0.59±0.04 | 0.61±0.05 | 0.54±0.03 |
| $c_1=0$ | $c_2=-0.5$ | 0.61±0.00 | 0.53±0.03 | 0.50±0.01 | 0.48±0.02 |
| $c_1=0$ | $c_2=0.0$ | 0.54±0.02 | 0.40±0.09 | 0.39±0.07 | 0.43±0.05 |

Figure 3: **Full sweep** over the generalized action spaces with 100 different parameterizations ($c_4 = 0$). We report the mean and standard error of the normalized return across three seeds for each table entry, each representing one unique combination of the generalized action space.

**Robot Platforms**   To evaluate the choice of action spaces, we use two robots in our case study: The Unitree Go1 and the evoBOT. The Unitree Go1 is a quadrupedal robot with 12 electric actuators standing 40 cm tall and weighing 12 kg. The evoBOT is a dynamically unstable robot based on the principle of a compound inverted pendulum containing 8 electric actuators, standing 80 cm high (without the arms) and a total weight of 50 kg.

**Sim-to-Real**   To transfer the trained policies from simulation to the real world, we use standard domain randomization techniques. To model uncertainty in the robot's sensing and actuation, we add Gaussian noise to the sensor observations and implement a lag model to simulate processing delays. To make the robot robust to varied terrains and disturbances, we also randomize the ground friction randomization and randomly push the robot during training. The details of the domain randomization parameters and ranges for each task are given in Section 6.

## 4   Results

In this section, we evaluate our research questions through experiments in simulation. To verify that our environment is reflective of a sim-to-real scenario, we also transferred the highest-performing learned policies from Section 4.1 to the Go1 and evoBOT robots in the real world.

### 4.1   How does the choice of action space impact learning?

We train policies for the Go1 and evoBOT covering 100 sets of parameters $c_1, c_2, c_3$ in the generalized action space parameterization (Section 3). All policies are trained in simulation. The results are reported in Fig. 3. Our aim in this experiment was not to find the most optimal action space for each task, but to broadly characterize the trends in the action space parameters for two robots with different dynamics. To accomplish this, it wasn't important to use the most efficient search method (e.g. bayesian optimization) but instead cover a wide variety of action spaces to expose trends. Therefore, we opted for a simple grid search. The ranges for the grid search were chosen to span the settings provided in OmniIsaacGymEnvs as the tuned default action spaces for similar robots.

First, we observe that the *optimal action space can strongly depend on the learning task*. Even for locomotion environments, the Go1 and evoBOT have almost opposite trends in terms of how adjusting the action space parameters impacts their performance.

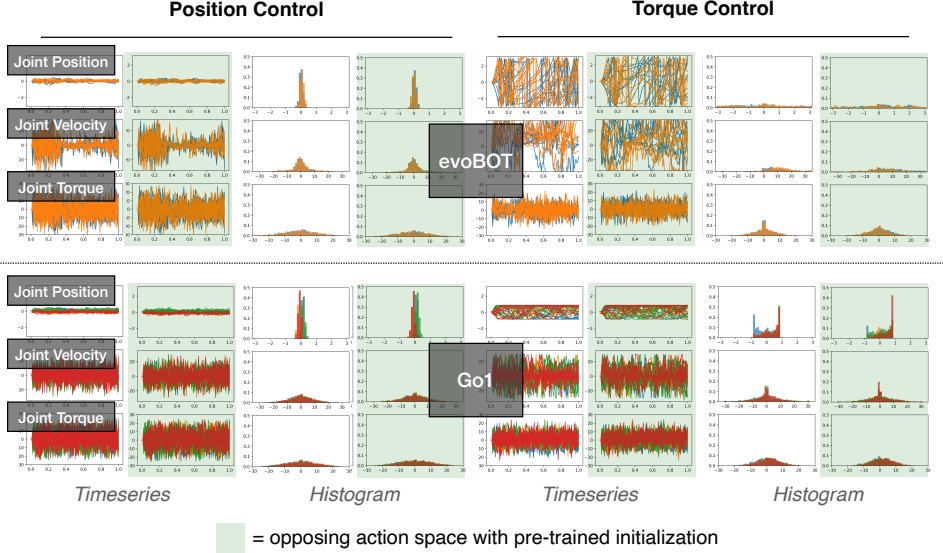

**Position Control**    **Torque Control**

*Timeseries*    *Histogram*    *Timeseries*    *Histogram*

= opposing action space with pre-trained initialization

Figure 4: **Random rollout behavior** of different action spaces. For each action space and robot, the left column plots the joint positions, velocities, and torques during the rollout of a randomly initialized policy, while the right column plots the histogram of the same states. The different colored lines in each plot correspond to different joints of the robot.

Second, we see that the *learning dynamics can be very sensitive to the choice of action space parameters*. For evoBOT, introducing nonzero $c_1$ corresponding to a position control loop substantially weakens performance, as does increasing $c_2$ corresponding to the damping parameter. For Go1, the robot cannot achieve a high reward without a sufficiently high $c_1$. The difference is due to a difference in each robot's natural dynamics. The Go1 benefits from a bias towards a nominal standing pose since learning gravity compensation is a key part of its task and the optimal strategy involves oscillating the joints around their nominal pose. The evoBOT on the other hand requires continuous rotations of the wheel to locomote so the bias towards a nominal position introduced by nonzero c1 can harm performance.

### 4.2 Is action space selection mostly about tuning initial exploration?

Different action spaces yield different initial exploratory behavior, illustrated in Fig. 4, which plots the joint position, velocity, and acceleration during random rollouts. The initial behavior is thought to be impactful for learning some tasks; for example, Hwangbo et al. [1] observed that for quadruped locomotion, the position control action space results in standing behavior under common network initializations while commanding torques directly will bias the robot to wriggle and fall. To experimentally evaluate the impact of the initialization, we propose an intervention to alter the initial exploration of the policy. We randomly initialize teacher policy $\pi_{\theta_T}^{c_1,c_2,c_3,c_4}$ and student policy $\pi_{\theta_S}^{c_1,c_2,c_3,c_4}$. The student policy is trained using supervised learning to imitate the randomly initialized teacher's torque outputs but with a different internal action space: $\min_{\theta_S} E[(\pi_{\theta_T}^{c_1,c_2,c_3,c_4}(s) - \pi_{\theta_S}^{c_1,c_2,c_3,c_4}(s))^2]$, for $s \sim \pi_{\theta_T}^{c_1,c_2,c_3,c_4}(s)$ obtained from rolling out the teacher in the training environment. This will incentivize the rollouts of the student operating in one action space to match the initial exploratory behavior of another. The green highlighted blocks in Fig. 4 show that the random rollouts from the student policy adopt similar characteristics to each teacher.

To evaluate the impact of initial exploration on the dynamics of learning, we evaluate four configurations in Fig. 5: Torque from Scratch, Torque with PD-like Initialization, Position from Scratch, and Position with Torque-like Initialization, where $X$ with $Y$-like initialization means we pretrain the $X$ policy to imitate rollouts from a randomly initialized $Y$ policy, then train the $X$ policy with reinforce-

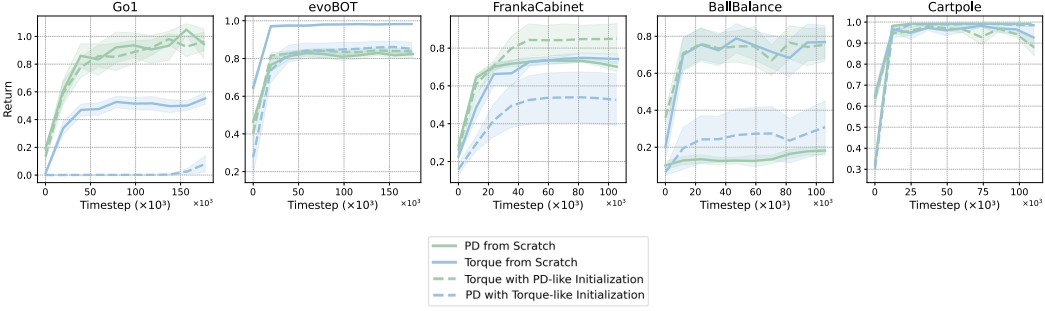

Figure 5: **Initial exploratory behavior** plays a major role in the performance of different action spaces. For dashed lines, the policy underwent a pretraining stage where it was trained through imitation learning to match the initial random rollouts sampled from another action space. Mean and standard error of normalized returns are reported across three random seeds.

ment learning. For Go1, torque policy with position-like initialization boosts the final performance close to training with a position-controlled policy. Conversely, for evoBOT where the torque action space is more performant, pre-initializing a PD controller with torque-like initialization results in a gain, although not a full recovery of final performance. We also consider the FrankaCabinet, Ball-Balance, and Cartpole tasks from the OmniIsaacGymEnvs suite. We find that for some tasks (Go1, evoBOT), the performance gap between action spaces can be partly or fully recovered by intervening in the initial exploration behavior, while for others, this aspect is less influential. Surprisingly, FrankaCabinet obtains the highest performance from a torque policy with PD-like initialization, suggesting it may benefit from initially exploring like a PD policy and later exploring like a torque policy. This seems intuitively reasonable since the Franka can benefit from gravity compensation at the start derived from the bias of PD control and later explore a different space of movements to open the drawer.

### 4.3 Is action space selection mostly about expressive capacity?

The complementary view of action spaces being tools for exploration would be that they have different expressive capacities due to their different architecture. We find this is not true for policies that run at synchronized frequency, and we can train a policy with any of position, delta position, or torque action space to match a high-performing policy trained in another action space using distillation (see Table 2). This is naively true because neural networks are universal function approximators; however, it illustrates that our comparison is devoid of factors that would make the policies impossible to imitate, such as varied behavior between timesteps, which we evaluate in Section 4.4.

### 4.4 How important is the policy behavior between timesteps?

We examine the impact of temporal aspects on policy performance across different action spaces (see Fig. 6). Policies are trained for evoBOT and Go1 with control frequencies ranging from 25 to 200 Hz, and their performance is evaluated based on total reward. All experiments maintain a fixed physics frequency of 200 Hz and implement the corresponding RL frequency via frame skips of $[1, 2, 4, 8]$. Results indicate that control frequency significantly affects performance, with higher frequencies generally yielding better rewards in torque control mode for evoBOT, while lower frequencies also perform well for Go1.

Additionally, the behavior between physics substeps impacts final policy performance. We analyze scenarios where torques are recomputed at each physics substep ($f^{\text{Ctrl}} = f^{\text{Phys}}$) versus when they remain fixed until the next RL update ($f^{\text{Ctrl}} = f^{\text{RL}}$). Our findings suggest that updating torques at the physics level leads to superior performance for most action spaces and tasks, especially in position control, likely due to the effects of stiffness and damping in PD control.

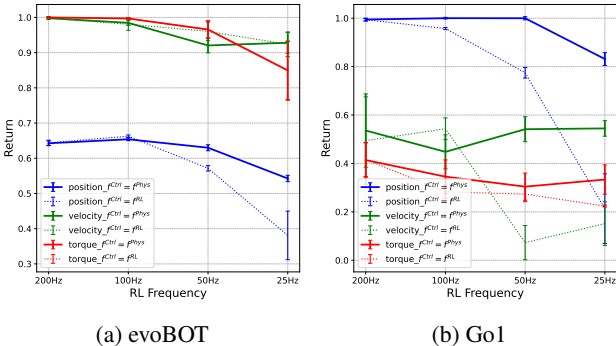

|              | Normalized Return |              |
|--------------|-------------------|--------------|
|              | evoBOT            | Go1          |
| *Teacher*          | $1.00 \pm 0.00$ | $1.00 \pm 0.07$ |
| Position Student   | $1.00 \pm 0.00$ | $1.00 \pm 0.13$ |
| Velocity Student   | $1.00 \pm 0.00$ | $1.03 \pm 0.16$ |
| Torque Student     | $1.00 \pm 0.00$ | $0.99 \pm 0.18$ |

|                |                |
|----------------|----------------|
| (a) evoBOT     | (b) Go1        |

Figure 6: **Temporal behavior** of RL frequencies and torque updates significantly influences policy performance across action spaces. Performance is evaluated across three seeds, with mean and standard error of normalized returns.

Table 2: **Expressive capacity** plays a negligible role in the performance of different action spaces. The best policy from the action space sweep serves as the teacher, while student policies imitate it using specific action representations. Mean and standard error of normalized returns are reported across three random seeds.

## 5 Practical Guidelines for Action Space Selection

While our findings generally support that the choice of action space is specific to the robot and task, we distill some overall guidelines to assist practitioners when selecting action space for new tasks:

1. When selecting an action space for learning, first consider the robot's dynamics and the types of movements you expect to be required. If keeping the joints close to a nominal posture requires some torque, a position gain may be helpful. If the joints require continuous rotation, consider a torque action space.

2. The choice of action space has a considerable impact on the initial exploratory behavior of the policy. You can gain intuition for this by visualizing the position, velocity, and torque of random rollouts. Combining the initial behavior of one action space with the representation of another is possible through an initial imitation stage.

3. In general, select the action space gains that provide the most advantageous learning dynamics for your task. If you need to deploy the final policy using a different set of gains, you can use teacher-student learning to perform the conversion.

4. Policy behavior between timesteps can influence performance, but with mixed trends. It's generally preferable to run the policy at a higher frequency and without any frame skip.

## 6 Limitations and Future Work

Although our work identified trends in action space design across multiple tasks, these trends do not fully explain the differences in learning dynamics. For example, initializing the policy to mimic the initial exploration of another action space improved performance in several tasks but did not yield identical learning curves, indicating that some factors remain unexplained. While we enhanced our mechanistic understanding of how action space selection affects learning, we could not leverage this understanding to stabilize training or introduce a new initialization scheme that consistently improves performance across tasks. This underscores the challenge of creating robust training that is less sensitive to environment design, as action spaces introduce strong, task-specific biases.

Future work could focus on establishing metrics to predict whether an action space is suitable for new tasks *before* training, potentially accelerating automated environment design, such as using LLMs [28]. Additionally, it is important to note that our analysis was restricted to the behavior of action spaces in combination with the PPO algorithm. While we believe our conclusions are useful since PPO is widely adopted for sim-to-real tasks, further research is needed to explore how these results generalize to other RL algorithms.

**Acknowledgments**

This research has received funding from the Federal Ministry of Education and Research of Germany and the state of North Rhine Westphalia as part of the Lamarr-Institute for Machine Learning and Artificial Intelligence (LAMARR22B) and the Junior Research Group DynaFoRo (01IS22074). This research was partly supported by Hyundai Motor Company, the MIT-IBM Watson AI Lab, and the National Science Foundation under Cooperative Agreement PHY-2019786 (The NSF AI Institute for Artificial Intelligence and Fundamental Interactions, http://iaifi.org/).

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

# A Training Details

| Hyperparameter | Value |
|---:|:---|
| # timesteps per rollout | 48 |
| # minibatches per epoch | 5 |
| mini-batch size | 32768 |
| discount factor | 0.99 |
| GAE parameter | 0.95 |
| learning rate | adaptive [29] |
| kl threshold | 0.008 |
| # workers | 1 |
| # environments per worker | 4096 |

Table 3: Hyperparameters used during training with PPO [26].

| Hyperparameter | Value |
|:---|:---|
| Hidden layer dimensions | $[512, 256, 128]$ |
| Activation function | `elu` |

Table 4: Neural network architecture for the Go1 quadruped locomotion task.

| Hyperparameter | Value |
|:---|:---|
| Hidden layer dimensions | $[128, 64]$ |
| Activation function | `elu` |

Table 5: Neural network architecture for the evoBOT locomotion task.

| Term | Form | Coefficient |
|---:|:---|:---|
| Lin vel tracking (xy) | $\exp\{-\lvert \mathbf{v}_{xy,t} - \mathbf{v}_{xy,t}^{\mathrm{cmd}}\rvert^2/\sigma_{vxy}\}$ | 1.0 |
| Ang vel tracking (yaw) | $\exp\{-(\omega_{z,t} - \omega_{z,t}^{\mathrm{cmd}})^2/\sigma_{\omega z}\}$ | 0.5 |
| Joint acceleration | $\lvert \ddot{\mathbf{q}}_t \rvert^2$ | $-2.5e-7$ |
| Delta torque | $\lvert \tau_t - \tau_{t-1}\rvert^2$ | $-5e-8$ |
| Delta pos | $\lvert \mathbf{q}_t - \mathbf{q}_{t-1}\rvert^2$ | $-4e-3$ |
| Lin vel (z) | $\mathbf{v}_z^2$ | $-1.0$ |
| Ang vel (roll-pitch) | $\lvert \omega_{xy}\rvert^2$ | $-0.05$ |
| Power | $\lvert \tau_t \cdot \dot{\mathbf{q}}_t \rvert^2$ | $-2e-5$ |
| Foot clearance | $\sum_{\mathrm{foot}}(h_{z,\mathrm{foot}}^f - h_z^{f,\mathrm{cmd}})^2 C_{\mathrm{foot}}^{\mathrm{cmd}}(\theta^{\mathrm{cmd}},t)$ | $-20.0$ |
| Base height | $(h_z - h_{\mathrm{ref}})^2$ | $-80.0$ |
| Base orientation | $\lvert g_t\rvert^2$ | $-0.2$ |
| Foot contact schedule (force) | $\sum_{\mathrm{foot}}[1 - C_{\mathrm{foot}}^{\mathrm{cmd}}(\theta^{\mathrm{cmd}},t)]\exp\{-\lvert \mathbf{f}^{\mathrm{foot}}\rvert^2/\sigma_{cf}\}$ | 0.2 |
| Foot contact schedule (velocity) | $\sum_{\mathrm{foot}}[C_{\mathrm{foot}}^{\mathrm{cmd}}(\theta^{\mathrm{cmd}},t)]\exp\{-\lvert \mathbf{v}_{xy}^{\mathrm{foot}}\rvert^2/\sigma_{cv}\}$ | 0.2 |
| Raibert heuristic | $(\mathbf{p}_{x,y,\mathrm{foot}}^f - \mathbf{p}_{x,y,\mathrm{foot}}^{f,\mathrm{cmd}})^2$ | $-5.0$ |

Table 6: Reward for quadruped locomotion (adapted from [30] to eliminate dependence on action representation, e.g. action change penalty is replaced by other smoothness terms).

| Term | Form | Coefficient |
|---:|:---|:---|
| Lin vel tracking (x) | $\exp\{-\lvert \mathbf{v}_{xy,t} - \mathbf{v}_{xy,t}^{\mathrm{cmd}}\rvert^2/\sigma_{vxy}\}$ | 1.0 |
| Ang vel tracking (yaw) | $\exp\{-(\omega_{z,t} - \omega_{z,t}^{\mathrm{cmd}})^2/\sigma_{\omega z}\}$ | 0.5 |
| Joint acceleration | $\lvert \ddot{\mathbf{q}}_t\rvert^2$ | $-1e-4$ |
| Delta torque | $\lvert \tau_t - \tau_{t-1}\rvert^2$ | $-1e-4$ |

Table 7: Reward for evoBOT locomotion (adapted from [6] for improved velocity tracking and to eliminate dependence on action representation, e.g. action change penalty is replaced by other smoothness terms).

| Term | Dim | Scale | Noise Std |
|---|---|---|---|
| Gravity vector | 3 | 1.0 | 0.01 |
| Velocity command | 3 | 1.0 | 0.0 |
| Joint positions | 12 | 0.5 | 0.005 |
| Joint velocities | 12 | 0.05 | 1.0 |
| Clock variable | 1 | 1.0 | 0.0 |

Table 8: Observations for the Go1 quadruped locomotion task.

| Term | Dim | Scale | Noise Std |
|---|---|---|---|
| Velocity command | 2 | 2.0 | 0.0 |
| Linear x-velocity | 1 | 2.0 | 0.0003 |
| Angular y/z-velocity | 2 | 0.25 | 0.0005 |
| Pitch angle | 1 | 0.5 | 0.005 |
| Joint positions | 2 | 0.5 | 0.001 |
| Joint velocities | 2 | 0.05 | 0.01 |

Table 9: Observations for the evoBOT locomotion task.

| Term | Min | Max |
|---|---|---|
| Ground friction | 0.4 | 2.0 |
| Ground restitution | 0.0 | 1.0 |
| Initial joint velocities | −0.1 | 0.1 |
| Random pushes (every 10 seconds, m/s) | −0.5 | 0.5 |

Table 10: Domain randomization parameters for the Go1 quadruped locomotion task.

| Term | Min | Max |
|---|---|---|
| Ground friction | 0.5 | 1.0 |
| Ground restitution | 0.0 | 1.0 |
| Initial joint velocities | −0.1 | 0.1 |
| Random pushes (every 4 seconds, m/s) | −0.5 | 0.5 |

Table 11: Domain randomization parameters for the evoBOT locomotion task.

