# OpenReview forum: "Action Space Design in Reinforcement Learning for Robot Motor Skills"
_robot-learning.org/CoRL/2024/Conference — CoRL 2024_

### Official Review · Reviewer_29AA · 2024-07-18
**Interesting paper, but needs clarification**

**Originality:** 3
**Technical Quality:** 3
**Clarity Of Presentation:** 3
**Potential Impact:** 2
**Recommendation:** 3
**Confidence:** 4

**Review:**

The paper tackles an interesting problem and the unified formulation to describe the different action spaces is a good contribution.
Overall, the paper goes in the right direction but lack clarity on some points and needs more polishing.

For example, it should be more explicitly mentioned that it considers only "configuration" based action space (no Cartesian/impedance control).
It also seems that Table 1 (and the generalized framework) is only valid when the policy network and low-level control are running at the same frequency (d=1).

It is not clear if all the results with evoBot and Go1 quadruped are in simulation or on real robots (or both?).
(I detail my different questions/remarks to improve clarity in the next section "questions for rebuttal")

In the results, no uncertainty is reported, which should be the case at least for simulation.

Unfortunately and as noted by the authors, the main takeaway message is that action space selection do have an impact on performance, but the paper provides no guideline to choose which action space to use in a given situation.

**Quality Of The Limitations Section:**

3

**Questions For Rebuttal:**

- In table 1, although it is clear how one-step action space can be retrieved, it is not clear for multi-step. It also seems that multi-step is also not clearly defined, but I assume it is when d > 1?
- I assume that every time that the generalized framework is used, only synchronous policy/low-level control is considered?

- It is not clear how the full sweep of Figure 3 was done. Was there only one run per entry?
 Is it simulation only? Why choose grid search (it is usually an inefficient way to search)?
 How was the search space chosen?
 Please report uncertainty when possible.

- For section 4.2, are the action space normalized? what is the original probability distribution used by PPO? (what happens if you replace PPO by SAC which uses a squashed Gaussian that should be less affected by action space limits)?

- For figure 5, random rollout corresponds to rollouts with randomly initialized policy?

- Figure 6/ section 4.4: only one run per datapoint? It would be good to normalize the performance (using random policy and max performance), because a difference of 0.002 in reward doesn't tell much if the performance changed a lot or not.
Again, please report uncertainty when possible.

- minor: typo in Fig 6: "no_udate" -> "no_update"

**Robotics Focus:**

4

**Summary Of Paper:**

The paper discusses the impact of action space selection on the performance of reinforcement learning for robot motor skills. Through a unified formulation, it explores the mechanisms by which action space influences performance, notably during the initial exploration phase and when using different control frequencies.

**Summary Of Recommendation:**

I recommend a weak rejection of the paper due to several issues that need to be addressed, including a lack of clarity on certain points and in the presentation of results. Overall, while the paper makes a good contribution by exploring the impact of action spaces, it needs further refinement before it can be considered for publication. **post-rebuttal update: I updated my score to weak accept but some additional points need to be addressed**

---

### Official Review · Reviewer_15bq · 2024-07-21
**Important topic; concerns around generalizability, limited scope and insufficient runs**

**Originality:** 2
**Technical Quality:** 2
**Clarity Of Presentation:** 3
**Potential Impact:** 2
**Recommendation:** 2
**Confidence:** 4

**Review:**

Strengths:

- The paper addresses an important and often overlooked topic, as the selection of action space in previous research has often been ad hoc.
- It introduces a general parameterization of action space, enabling the representation and comparison of common action spaces within a unified framework.
- Experiments are conducted on both simulated and real-world robots.

Weaknesses:

- It is standard practice to report statistics from multiple runs in RL. However, the paper does not provide these details, and it seems that the results presented in the plots are based on a single run.
- The experiments only apply one algorithm, PPO, making it unclear whether the conclusions can generalize to other RL algorithms or are specific to PPO.
- The scope of this work is about “understanding” the impact of action space. But it lacks practical insights on “how“ to select action space for different tasks.
- The results in Figure 4 indicate a high dependency on the specific task, making it challenging to draw broad conclusions.
- The writing could be improved for better clarity, as detailed in "questions for rebuttal".

Minor:

- Typo on page 7, “instead RL-level” → “instead of …”
- Should the y-label in Fig.4 and 6 be return instead of reward?

**Quality Of The Limitations Section:**

2

**Questions For Rebuttal:**

- Table 2 caption mentions an “optimal teacher policy”: how was this policy obtained?
- As shown in Fig. 1, none of the three simulation environments use position-based action space. What’s the rationale of picking these environments?
- Line 160 refers to “The bottom row of Fig. 4”,  is this plot missing or incorrectly referenced?
- Line 175, mentions “exploring like a PD policy and later exploring like a torque policy”.
    - can you elaborate on what it means to be exploring like a PD policy, or exploring a torque policy?
    - In PPO, exploration typically reduce gradually. Is there more evidence to support the claim that the observed benefits come from exploration?
- In Sec. 4.4, Line 200 mentions performance increase from more frequent torque update. I wonder if this increase can be attributed to a more accurate physics simulation from the faster update, rather than the action space itself.
- I’m having trouble understanding the x-axis label “Decimation” in Fig. 6. Why is it not simply labeled “frame skip”?

**Robotics Focus:**

4

**Summary Of Paper:**

This paper studies the impact of action space in RL for robot control. It introduces a general reparameterization of the action space,  enabling the representation of common action space choices as a linear mapping from state and action to torque. Through experiments conducted in both simulated and real-world robots, the paper shows that action space design affects control performance in a task-dependent manner. Additionally, it demonstrates that performance difference is largely due to policy initialization and the interplay of control frequency and torque update frequency.

**Summary Of Recommendation:**

**Updated Post-Rebuttal**: The paper addresses an important topic, but the results are inconclusive despite that the authors have made some improvements during the rebuttal.

---

### Official Review · Reviewer_FMNG · 2024-07-27
**Review of paper**

**Originality:** 2
**Technical Quality:** 4
**Clarity Of Presentation:** 3
**Potential Impact:** 2
**Recommendation:** 3
**Confidence:** 4

**Review:**

Strengths
* The paper attempts to shed light on a very important aspect in robot learning - action space design. It is well known to be critical but it is rare to see research papers dedicated to the topic. Insights on this topic are likely to be very useful for the robot learning community.
* The introduction does a good job of motivating this research, and clearly lays out both the main questions the authors seek to answer and their main findings.
* I appreciated the generalized parameterization of action space formulation. This is original and insightful. It also enabled the authors to isolate the effect that action spaces have on exploration which was very interesting.
* Strong experimental design. The experiments are creative and insightful. They are effective at providing evidence to answer the main questions posed in the paper. I especially appreciated the analysis and discussion in sections 4.2 and 4.3.
* A thorough limitations section
* I appreciated the discussion in section 3.2 on temporal aspects of action space design

Weaknesses
* Lack of clarity and detail in multiple places. As-is I have a number of questions about the method, tasks, experimental setup and training parameters, and results (see questions for rebuttal). This is the core weakness of this paper.
* Narrow scope of action spaces considered. Whilst the authors consider multiple action spaces, they ignore a key choice of whether to choose an action space in end effector / leg space or joint space, and instead focus solely on joint space, without justification. For example, in manipulation end effector space remains a common choice.
* No real robot experiments, although the authors mention do provide a convincing argument that their results will hold in the real world (lines 218 - 219).
* Limited impact: Whilst this line of research has the potential to be very impactful, I think the paper as it stands is likely to have limited impact because there remain a number of open questions that limit the recommendations that the authors can give to those wishing to apply this work. Making progress on metrics that could be used to predict in advance of training whether an action space is likely to be useful (as discussed by the authors in lines 214-215) would be one way to increase the likely impact of this work.
* The related work is reasonably thorough, however the discussion of action spaces in lines 54-59 is missing a discussion of end-effector/leg space control vs joint space control (+ sub variants including delta-position control), as well as the choice of whether to remain in a continuous action space or to discretize. Additional choices such as heavily constraining action space ranges are also missing. Finally, I would have liked to see a discussion of how this work differs from prior work on action space design.

**Quality Of The Limitations Section:**

3

**Questions For Rebuttal:**

Related work
* How does this work differ to prior work on action space design?

Method
* Fig 2. h_t is undefined
* How do c1, c2, c3, and c4 in fig 2 relate to [c1, c2, c3, c4] in table 1?

Experimental setup
* What are the tasks that are being evaluated? What is the maximum and minimum reward per task?
* What are the policy training details - network architecture, number of training steps, number of training runs
* Sim-to-real - please give specific details. What parameters were domain randomized, what were the ranges, how much noise was added to the observations, how much latency?

Results
* 4.1
  * Line 139: What does a “full sweep” mean? I see that c1, c2 and c3 are varied but c4 appears constant. Is that the case and what was it’s value?
  * Was training in simulation or the real world?
  * What are your hypotheses about why there is such a difference between the Go1 and evoBOT embodiments?
* Fig 3: What does “maximum reward mean”? What is the maximum possible score?  Is the reported score a mean value? Over how many reported training runs? What was the variance of these results (RL is known to exhibit high variance)?
* Table 2: Is the reported score a mean value? Over how many training runs?
* Fig 5: What do the different color lines represent?
* Fig 6. Is there something wrong with the y axis of both charts? The value appears to be very low compared to Fig 3 and I assume this is the same task.
* How does the supplementary video relate to the paper? There are no real robot experiments in the paper and the video is of an evoBot wheeling around a space

Minor
* Line 31: This sentence doesn’t make sense
* Line 157: The teacher and student policy presumably have different action spaces, it would be helpful to make this explicit.
* Line 160: There is no bottom row to the referred figure
* Line 161: Should the be Fig 5 instead of Fig 4
* Line 196: effect → affect

**Robotics Focus:**

4

**Summary Of Paper:**

This paper assesses the role of action space design on the performance of robotic control policies trained using reinforcement learning. The work aims to understand if and how the choice of action space affects learning dynamics. Through a variety of simulated experiments, the work assesses the extent to which the action space affects initial exploration, a model’s expressive capacity, and the role of multi-timestep action formulations. The work concludes that the optimal action space is strongly dependent on the particular task, and that this can only be partially explained by differences in exploration.

**Summary Of Recommendation:**

This paper is on an interesting and relevant topic, with nicely designed experiments, but poor clarity as the paper stands. Additionally, the action space scope is somewhat narrow, ignoring a key choice between end effector / leg space and joint space control.  The main reasons for my recommendation are (a) clarity which could be addressed during rebuttal and (b) narrow action space scope. Given there is no algorithmic, dataset, or system design novelty and the experimental results are in simulation, I wish to see a deeper exploration of action spaces and more insight, for example by including in this work the metric proposed in the limitations section.  **Post-rebuttal update: The revisions provided during rebuttal substantially improved the paper. As a result I am revising my recommendation to weak accept**

---

### Author Rebuttal · Authors · 2024-08-12

The zip file contains the improved manuscript with changes from the rebuttal highlighted in blue.

---

### Decision · Program_Chairs · 2024-09-04

**Decision:**

Accept

**Comment:**

Strengths
- Highly relevant topic for all robot RL researchers
- Strong experimental design
- Nice discussion and limitations sections

Weaknesses
- Many details unclear
- Unclear if experiments report multiple runs
- Evaluating on multiple RL algorithms would have been nice to show generality
- Scope of action space a bit limited and also the related discussion needs to be improved
- Real robot experiments would strengthen the paper
- Unclear how to apply lessons learned in practice

# After Discussion Phase
All reviewers highly appreciated the additional information and improved experiments. Overall that improved the paper significantly. Reviewer 29AA still has 2 concerns that are straightforward to fix. Reviewer FMNG still maintains 2 suggestions on how to further strengthen the paper. While I agree with Reviewer 15bq that the results aren't as conclusive as one might have hoped, this is unfortunately typical for RL (whether a methods performs well or not is often highly task specific), and I believe the insights and new practical guidelines are a valuable contribution.